# Online Monitoring of Catalytic Processes by Fiber-Enhanced Raman Spectroscopy

**DOI:** 10.3390/s24237501

**Published:** 2024-11-25

**Authors:** John T. Kelly, Christopher J. Koch, Robert Lascola, Tyler Guin

**Affiliations:** Savannah River National Laboratory, 301 Gateway Drive, Aiken, SC 29803, USArobert.lascola@srnl.doe.gov (R.L.); tyler.guin@srnl.doe.gov (T.G.)

**Keywords:** Raman spectroscopy, fiber enhancement, hydrogen gas, catalytic reaction monitoring, hollow waveguide

## Abstract

An innovative solution for real-time monitoring of reactions within confined spaces, optimized for Raman spectroscopy applications, is presented. This approach involves the utilization of a hollow-core waveguide configured as a compact flow cell, serving both as a conduit for Raman excitation and scattering and seamlessly integrating into the effluent stream of a cracking catalytic reactor. The analytical technique, encompassing device and optical design, ensures robustness, compactness, and cost-effectiveness for implementation into process facilities. Notably, the modularity of the approach empowers customization for diverse gas monitoring needs, as it readily adapts to the specific requirements of various sensing scenarios. As a proof of concept, the efficacy of a spectroscopic approach is shown by monitoring two catalytic processes: CO_2_ methanation (CO_2_ + 4H_2_ → CH_4_ + 2H_2_O) and ammonia cracking (2NH_3_ → N_2_ + 3H_2_). Leveraging chemometric data processing techniques, spectral signatures of the individual components involved in these reactions are effectively disentangled and the results are compared to mass spectrometry data. This robust methodology underscores the versatility and reliability of this monitoring system in complex chemical environments.

## 1. Introduction

Recognized for its broad applicability, Raman spectroscopy offers significant advantages in terms of specificity, speed, accuracy, simplicity, and stability. One of the major weaknesses of the analytical approach is the inherent weakness of Raman scattering, resulting in poor sensitivity. However, there are several techniques that enhance the Raman signal (e.g., surface plasmons, resonance enhancement, and multi-pass/cavity-based techniques) and there are systematic improvements that have been recently demonstrated and developed [1,2,3,4]. In the 1970s, thin film-coated optical fibers were demonstrated as a waveguide enhancement approach for liquid samples [5]. However, recent work in fiber-based approaches has expanded as a sensor for gaseous species (i.e., CO, CO_2_, H_2_, D_2_, N_2_, O_2_, SO_2_, NH_3_, CH_4_, and other small hydrocarbons) [6,7,8,9,10,11,12,13].

The first of two catalytic processes monitored here is carbon dioxide methanation (CO_2_ + 4H_2_ → CH_4_ + 2H_2_O) in which the spectral features of each component have been reported to navigate performance conditions [14,15,16]. Analytical techniques commonly used for monitoring gas-phase reactions include mass spectrometry and infrared spectroscopy (diffuse reflectance infrared Fourier transform spectroscopy or DRIFTS) [17,18]. For this reaction, features in mass spectra can be compromised by mass coincidence (N_2_ and CO have m/z ~28) and inactive vibrations in the infrared spectra prevent detection of H_2_ and N_2_. In contrast, Raman spectroscopy provides a single analytical approach to track all the involved gases simultaneously and thus more completely assess the performance of the catalyst in real time [11]. Flow rates, temperatures, and molar ratios are all modified to determine key kinetic properties. Robust data acquisition is required to adjust these parameters for optimal catalytic performance.

The second of two catalytic processes demonstrated is ammonia cracking or the chemical process where NH_3_ is decomposed into its diatomic constituent elements, N_2_ and H_2_ [19,20,21,22]. The most common cracking techniques rely on catalyst iron oxide (Fe_3_O_4_) [23]; however, catalytic efficiency can be further optimized by introducing elements like potassium or aluminum for synthesis and incorporating nickel or ruthenium for decomposition [24,25,26]. Monitoring reactants and products in real time plays a role in computing reaction rates in relation to the composition of the catalyst under consideration. Raman spectroscopy provides a noninvasive approach for observing relative and absolute concentrations for samples in gas, liquid or solid forms and is ideal for interrogating small molecular species [27,28].

In this work, two catalytic processes are monitored via real-time Raman spectroscopy for the activity of catalysts on Al_2_O_3_ support [29,30]. The primary focus of this work is the demonstration of real-time monitoring of gas cracking and synthesis by Raman spectroscopy. The primary aim is to demonstrate the analytical capabilities of this experimental design and gain insight into the feasibility of quantitative measurements utilizing a hollow-core waveguide as a flow cell in addition to the signal enhancement [31].

## 2. Materials and Methods

Experiments for monitoring HD by mixing H_2_ and D_2_ across platinum-coated alumina pellets have been previously reported in the literature by Telle and coworkers. In brief, Raman spectra are collected by flowing gases through a hollow-core waveguide [11]. A backward Raman configuration is used for the analysis of gaseous samples flowing through a waveguide to accomplish enhanced collection of Raman scattering. Raman spectroscopic measurements were carried out with a Horiba iHR320 spectrometer (0.318 m; 600 lines/mm grating) equipped with a Syncerity cooled CCD detector (–60 °C) [32]. Spectra were collected in the 180° backscattering geometry using 532 nm excitation (532 nm 1500 mW Green DPSS, Civil Laser Supplier. NaKu Technology Co., Ltd., Hangzhou, China). The reflective waveguide is a commercially available silver-lined, glass capillary with an inner diameter of 1 mm and a length of 500 mm (Guiding Photonics, Torrance, CA, USA). The waveguide is held in nested stainless-steel tubes with matched inner and outer diameters. These tubes support the waveguide and facilitate connection to the gas handling systems. Restricted flow outside the tubes, due to the matched diameters, ensures gas flow through the waveguide [33].

Stainless-steel pinhole (800 micron) endcaps cover the endcaps of the waveguide to remove background scattering from the annular silica surface that would otherwise be illuminated. Data were collected and presented in real time (Horiba LabSpec6 software, Solo Model Exporter 9.2.1, PLS toolbox, Eigenvector Research) to control the detector, grating position, and acquire data. Data analysis was performed offline using a chemometric pre-processing software package (PLS toolbox, Eigenvector Research, Wenatchee, DC, USA) for background subtraction, smoothing, and peak fitting. Optical lenses and mounts were purchased through Thorlabs Inc., including a narrowband pass filter, 532 nm dichroic filter, mounts, posts, silver mirrors and focusing lenses. Figure 1 shows a schematic with optical elements for sampling and collection of Raman spectra.

For CO_2_ methanation reactions, the catalytic reactor bed was packed with 500 mg of a nickel-based (Ni/Yb/K/Al_2_O_3_) catalyst and was activated under a 20% H_2_ (10 mL min^−1^ H_2_ and 40 mL min^−1^ Ar) atmosphere at 600 °C for three hours. The flow was then purged of H_2_ by flowing pure Ar for 30 min before reagent gases were added to ensure any H_2_ detected was that from CO_2_ methanation. Then, the reaction took place under varying amounts of CO_2_ in H_2_ at 450 °C. CO_2_ (>99.98) was obtained from Sigma Aldrich (St. Louis, MO, USA). Nickel nitrate (99.9%), ytterbium nitrate (99.9%), and potassium nitrate (99.9%) were all obtained from ThermoFisher (Waltham, MA, USA) and utilized for the catalyst synthesis without further purification. γ-Al_2_O_3_ was obtained from Sasol (Johannesburg, South Africa) (lot# TK2347) and used without further purification.

For ammonia cracking, the catalytic reactor bed was packed with 500 mg of a trimetallic ruthenium (Ru/Y/K/Al_2_O_3_) catalyst and was activated under a 20% H_2_ (10 mL min^−1^ H_2_ and 40 mL min^−1^ Ar) atmosphere at 450 °C for three hours. The flow was then purged of H_2_ by flowing pure Ar for 30 min before ammonia was added to ensure any H_2_ detected was that from ammonia decomposition. Then, the reaction took place under either 10% NH_3_ in Ar or 100% NH_3_ atmospheres at temperatures ranging from 250 to 450 °C. NH_3_ (>99.98) was obtained from Sigma Aldrich. The exhaust was connected to the Raman spectrometer and a mass spectrometer (Cirrus 2, MKS, Andover, MA, USA). Ruthenium nitrosylnitrate (99.9%) and yttrium nitrate (99.9%) were both obtained from ThermoFisher.

## 3. Results and Discussion

While catalysts are developed based on efficiency, performance, deactivation, cost of materials, and effectiveness, the analytical measurement technique for each of these criteria has yet to be standardized across the community. The results shown here demonstrate the versatility of Raman spectroscopy in CO_2_ methanation for detecting reactants, products, and environmental components. Figure 2 is a Raman spectrum of the effluent from a reactor with CO_2_ and H_2_ flowed at optimized ratios (concentrations) and speeds in the presence of a nickel-based catalyst. The CO_2_ features centered at 1286 and 1388 cm^−1^ and CO feature at 2144 cm^−1^ have been previously reported and are used to monitor effective catalysis (CO_2_ conversion) and breakthrough. The H_2_ features at 354, 587, 814, 1035 and 4161 cm^−1^ are also in agreement with previously reported values and provide insight into the stoichiometric conditions of the catalyst. The dominating methane features at 2915 cm^−1^ allow for methane to be detected under poor conditions for catalysis.

The temporal selectivity for CO_2_ methanation can be best represented through extracting the reactants and products of the reaction using multivariate curve resolution (MCR). Here, Raman spectra of known concentrations as well as sample data are batch processed by baseline subtraction and signal averaging. The known concentrations serve as an external calibration with a minor nitrogen feature at 2330 cm^−1^ serving as the internal standard. Figure 3 is separated into six distinct regions of interest and shows the three major components from the MCR loading plots generated from 120 Raman spectra. The CH_4_ component (blue) is the product of the reaction and is monitored along with the reactants (H_2_ in green and CO_2_ in black). In Regions A, B, and C, the flow rates of H_2_ and CO_2_ are recorded at 50:0, 0:50, 50:0 mL min^−1^, respectively. Low amounts of methane were observed in Region B, most likely due to lingering hydrogen from the catalyst activation reacting to form CH_4_. The flow rates in Region D are 50:20 and continuous CO_2_ methanation is observed. Region E shows the increase in H_2_ flow from 100 to 150 to 200 mL min^−1^ and shows that the additional hydrogen flow improves the conversion of the catalyst as CO_2_ decreases. Lastly, Region F shows that increasing the CO_2_ flow from 20 to 30 to 50 mL min^−1^ decreases the rate of reaction. Additionally, it is important to note that as the CO_2_ increases, the H_2_ decreases due to its being consumed in the reaction. Thus, the Raman spectrometer can be utilized to identify products in the gas effluent stream and to track changes in the reactivity of the system in real time.

The Raman spectra of hydrogen isotopes have been previously reported by employing metal-coated hollow glass fibers in the application of passing H_2_ and D_2_ gases through a catalytic reactor to produce HD [11]. Here, a similar process is demonstrated known as ammonia cracking or 2NH_3_ → N_2_ + 3H_2_. Figure 4 (top) shows the Raman spectrum of 100% ammonia gas flowing through the catalytic reactor at 50 mL min^−1^ at a reactor temperature of 250 °C. The primary feature observed for ammonia is centered at 3337 cm^−1^. This is in agreement with previous reports [34], in which it has been characterized as the symmetric stretch (ν_s_). In addition to the primary feature, minor features include the out-of-plane bend (δ), the asymmetric stretch (ν_as_) and the asymmetric bending (ω).

The features observed for hydrogen and nitrogen have been previously reported in the spectral range of Figure 5 [35]. In addition to the fundamental Q-branches of the diatomic species (N_2_ 2330 cm^−1^ and H_2_ 4161 cm^−1^), there are rotational features observed across the spectral range from 100 to 4250 cm^−1^ [36,37]. Of these, the nitrogen rotational features are observed and can be compared to previously reported spectra in a similar Raman spectrum for ammonia to the nitrogen and hydrogen features in the spectral range of 100 to 600 cm^−1^. These features can be used as confirmatory or even primary indicators of process control at higher concentrations. Lewis and Houston previously reported these low-wavenumber features of the Raman spectrum of ammonia and assigned them to the ΔJ = ± 1 and 2 pure rotational states [38]. These results agree with the previously reported rotational constants for ammonia when considering the spectral range from 100 to 500 cm^−1^.

While some features overlap (completely or partially), standard chemometric approaches can be employed for deconvolution to include partial least-squares (PLS) modeling, first derivative (second-order polynomial) baseline correction, normalizing spectra to the area of a feature, and mean centering data to equally weight the data. Recent work by Lines and coworkers demonstrates the improvements by leveraging chemometric analyses for applications across multiple systems [39]. The combination of online monitoring, robust calibration methods and chemometrics provides a nondestructive analytical approach for gas processes, transfers, and separations [40,41]. This type of less common, spectroscopic approach can be directly compared to other analytical platforms (i.e., mass spectrometry).

The side-by-side comparison of Raman signals (counts at peak height) and selective ion monitoring signals (mass-to-charge ratio counts) is shown in Figure 6. These data were acquired with 10% ammonia in argon gas flowing through the catalytic reactor at a rate of 50 mL min^−1^. The initial temperature was 200 °C and the final temperature was 450 °C. The temperature was increased at a rate of 10 °C min^−1^. Both sets of data were processed in Eigenvector Research PLS-Toolbox (Solo+Model Exporter version 9.2). The Raman and mass spectra were processed with baseline corrections (Whittaker filter, λ = 100 and *p* = 0.001) and normalization of each of the largest features. Subtle differences in the curvature of the Raman and mass spectra are expected as a result of the heated transfer capillary and low flow to the mass spectrometer in addition to the location of the sampling position.

A major concern for Raman spectroscopy for online monitoring is providing defensible data in terms of intensity-to-concentration conversion. While canonical Raman experiments rely on stable laser power and spectrometer drift correction, fiber or waveguide enhancement techniques are significantly impacted by the sampling and collection efficiency. Minor misalignments for the laser excitation can significantly increase background scattering, and minor misalignment of the collection can decrease the observed Raman signal for a given concentration of gas. This work demonstrates the feasibility of monitoring reactive systems by waveguide-enhanced Raman scattering and does not aim to be a comprehensive study on quantifying limits of detection or other figures of merit for a validated analytical method. Further development of instrumental and procedural controls to facilitate operations by more quantitative approaches will lead to the employment of real-time monitoring by Raman spectroscopy.

## 4. Conclusions

Using a hollow-core waveguide for Raman spectroscopy provides significant scattering enhancement for gaseous samples flowing through the waveguide. While the advantages are obvious, there are several considerations to evaluate for future directions for this work to include sample compatibility, fluorescence, memory effects, and accessibility for alignment. Here, a Raman spectroscopy platform can provide real-time analytical support for catalytic reactions, specifically CO_2_ methanation and ammonia cracking. The low-wavenumber-region Raman spectrum of ammonia has been observed in this work but can be further explored for mixed isotope contributions. Compared to conventional Raman scattering geometries, the waveguide enhancement has been shown to provide sensitivity enhancements of at least 50× for hydrogen [11,32].

The chemometric techniques applied to the data support eventual implementation for real-time quantification of reactive species [42]. Chemometric tools were leveraged for post-processing data for calibration curves though mass flow-controlled dilution of target species in argon. A more acceptable approach for acquiring reliable data would be to devise a sample sequence that measures external calibration samples, quality control samples, and instrument blank samples. In addition to devising a reporting scheme for uploading to a laboratory information management system, a more robust approach for clustering and classifying unknown spectral features is projected for real-time analyses using Raman spectroscopy.

Here, a Raman spectroscopy platform is shown to provide real-time analytical support for catalytic reactions, specifically CO_2_ methanation and ammonia cracking. The low-wavenumber-region Raman spectrum of ammonia has been observed in this work but can be further explored for mixed isotope contributions. Future applications include tracking hydrogen isotope exchange in ammonia and methane, first in a “cold” environment using H_2_ and D_2_, then eventually in a radiological or “hot” glovebox in the Tritium Instrument Demonstration System at the Savannah River Site [43]. This specific application extends widely to radiological measurements to the accountancy in deuterium-tritium (DT) fusion energy as a non-invasive gas measurement technique [44]. An extra challenge to this work is that there is little literature of the vibrational spectroscopy of the multiple mixed isotopologue species for ammonia (four) and methane (five). The immediate next direction is focusing on generating these species using cracking methods, detecting the mixed isotopologues in real time, and using chemometric methods to extract the spectra of individual species from the inevitable mixtures that result. These results, confirmed by calculations of Raman intensity, will allow for an accurate concentration measurement of species that are difficult to isolate.

## Figures and Tables

**Figure 1 sensors-24-07501-f001:**
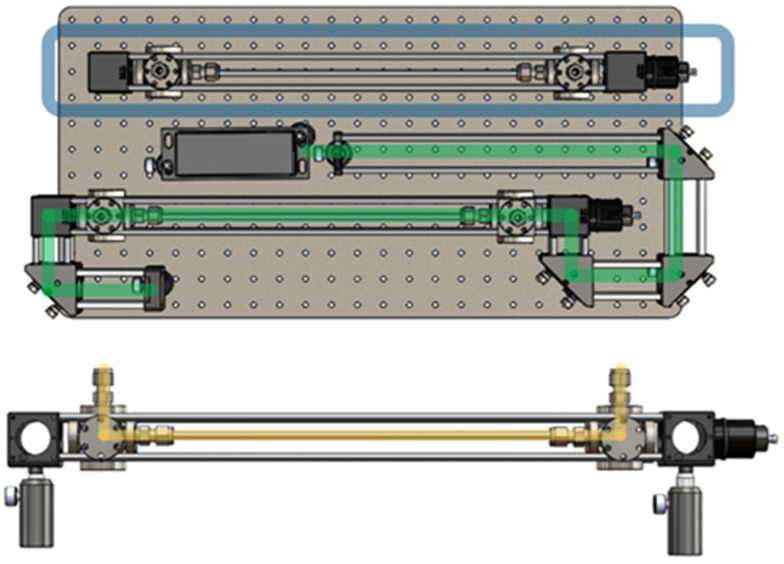
Experimental design of optical elements for sampling and collection of Raman spectra by workstation, (**top**) laser path in green and extra waveguide in blue. The (**bottom**) sample flow through the waveguide is highlighted in yellow.

**Figure 2 sensors-24-07501-f002:**
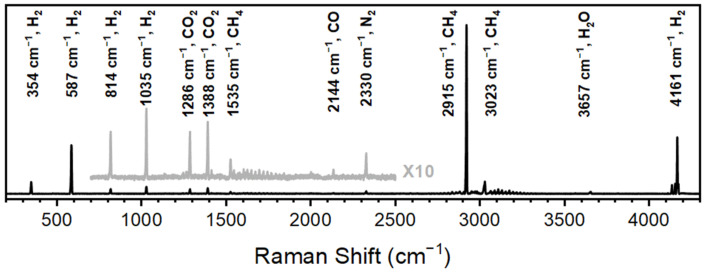
Raman spectra of CO_2_ methanation acquired over 10 s spectral accumulations.

**Figure 3 sensors-24-07501-f003:**
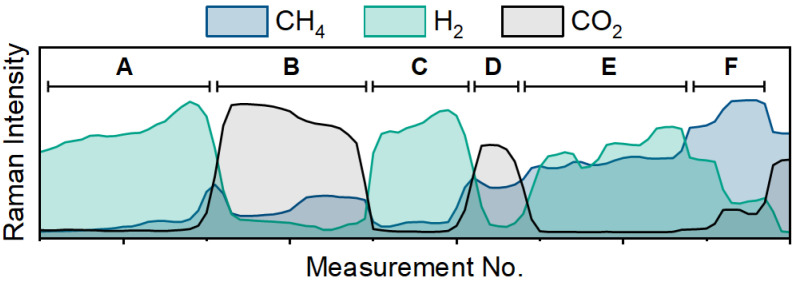
Traces from Raman spectra of CH_4_ (blue), H_2_ (green) and CO_2_ (black) with changes to the flow rate conditions of the feed gases to the reactor.

**Figure 4 sensors-24-07501-f004:**
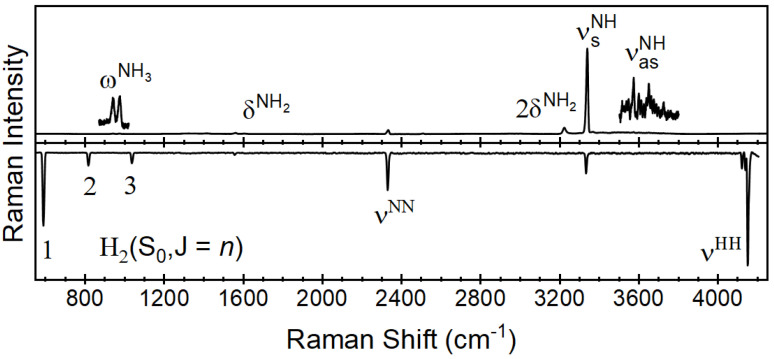
Raman spectra of ammonia reactant (**top**) and hydrogen/nitrogen product (**bottom**) flowing at 50 mL min and acquired over 10 s spectral accumulations.

**Figure 5 sensors-24-07501-f005:**
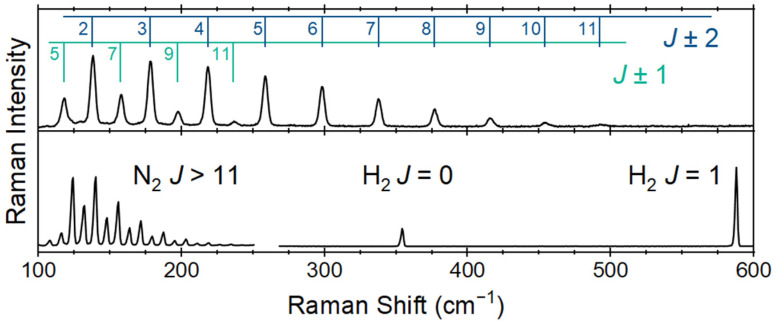
Low-wavenumber Raman spectra of ammonia (**top**) and hydrogen and nitrogen (**bottom**) flowing at 50 mL min and acquired over 30 s spectral accumulations.

**Figure 6 sensors-24-07501-f006:**
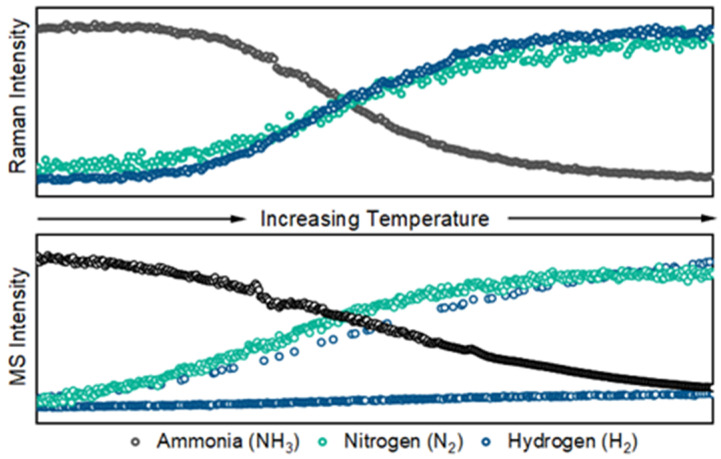
Peak heights of select Raman features of ammonia reactant (**top**) and hydrogen/nitrogen product (**bottom**) flowing at 50 mL min and acquired over 10 s spectral accumulations.

## Data Availability

Different catalysts were characterized and tested, including ruthenium on alumina and nickel on alumina (Ni/Al). Select catalysts on alumina may be the most compatible for use in tritiated environments. The specific synthesis is proprietary; however, the corresponding author can be contacted directly for additional information or discussion. In addition, inquiries on the detection limitations and sensitivity of the spectroscopic approach can also be sent to the corresponding author.

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
