# Peer review of "Online Monitoring of Catalytic Processes by Fiber-Enhanced Raman Spectroscopy"

_sensors, 2024, doi:10.3390/s24237501_

Round 1

Reviewer 1 Report

Comments and Suggestions for Authors

This article presents an innovative solution for real-time monitoring of reactions in enclosed spaces, optimized specifically for Raman spectroscopy applications. The research team used a hollow waveguide as a compact flow cell, serving not only as a channel for Raman excitation and scattering but also seamlessly integrating into the emission stream of a catalytic cracking reactor. This work demonstrates the versatility and reliability of the monitoring system in complex chemical environments.

1. The format of the paragraphs before and after Figure 1 is inconsistent with other formats in the entire text.

2. The font size in Figures 2 and 4 is too small. It is recommended to enlarge the font size.

3. It is recommended to place the image and text of Figure 6 on the same page.

Author Response

Comment 1: The format of the paragraphs before and after Figure 1 is inconsistent with other formats in the entire text.

Response 1: Thank you for pointing this out. The authors agree with the comment. Therefore, we have formatted the text of the paragraph before and after Figure 1.

Comment 2: The font size in Figures 2 and 4 is too small. It is recommended to enlarge the font size.

Response 2: Thank you for pointing this out. The authors agree with the comment. Therefore, we have formatted the font size in figures 2 and 4 so that they are easier to read.

Comment 3: It is recommended to place the image and text of Figure 6 on the same page.

Response 3: Thank you for pointing this out. The authors agree with the comment. Therefore, we have placed the image and text of figure 6 on the same page.

Reviewer 2 Report

Comments and Suggestions for Authors

This study reported that the innovative solution for real-time monitoring of reactions within confined spaces, optimized for Raman spectroscopy applications. The topic selection of the article is good, the manuscript is readability and rich workload. I suggest publishing on Sensors after minor revision. My comments are listed below:

1、Introduction: The content of the second paragraph should be adjusted to the last paragraph of the introduction.

2、There are some formatting errors in the text, such as the upper and lower corner marks (CO2, H2, cm-1).

3、The authors should compare other methods to enhance the catalytic process of Raman spectroscopy to highlight the advanced nature of this research.

4、Language quality should be further improved. For example, if there are many "We" in the text, the writer should avoid using the second person.

5、The author should add keywords such as "hollow waveguide" and "catalytic reaction monitoring" to better summarize the content of the article and enhance the retrieval of the article.

6、This study focuses on the application of ammonia cracking, and it is suggested to further explain the wide applicability of this method, such as its potential application in other reaction systems or its adaptability in different reaction environments.

7、It is suggested that the author further clarify the advantages of this technique compared with traditional Raman spectroscopy enhancement techniques (such as surface enhanced Raman scattering, SERS), especially in the aspects of sample complexity and sensitivity improvement.

8、In this study, it is mentioned that the chemometric model is used to quantify the mixed isotopes in real time, and it is suggested to add more details about the model construction.

Author Response

Comment 1: Introduction: The content of the second paragraph should be adjusted to the last paragraph of the introduction.

Response 1: Thank you for pointing this out. The authors agree with the comment. Therefore, we have adjusted the second paragraph of the intro to the last.

Comment 2: There are some formatting errors in the text, such as the upper and lower corner marks (CO2, H2, cm-1).

Response 2: Thank you for pointing this out. The authors agree with the comment. Therefore, we have corrected the subscripts and superscripts.

Comment 3: The authors should compare other methods to enhance the catalytic process of Raman spectroscopy to highlight the advanced nature of this research.

Response 3: Thank you for pointing this out. The authors agree with the comment. Therefore, we have included a sentence in the introduction that cites key reviews in enhancement technologies. While this Communication is only highlighting the concept of using a fiber as an improvement to online monitoring of ammonia cracking and CO2 methanation, a more comprehensive study on quantifying the enhancement factors is in preparation.

Comment 4: Language quality should be further improved. For example, if there are many "We" in the text, the writer should avoid using the second person.

Response 4: Thank you for pointing this out. The authors agree with the comment. Therefore, we have removed a majority of the pronouns throughout the text of the manuscript (both we and our).

Comment 5: The author should add keywords such as "hollow waveguide" and "catalytic reaction monitoring" to better summarize the content of the article and enhance the retrieval of the article.

Response 5: Thank you for pointing this out. The authors agree with the comment. Therefore, we have added these keywords.

Comment 6: This study focuses on the application of ammonia cracking, and it is suggested to further explain the wide applicability of this method, such as its potential application in other reaction systems or its adaptability in different reaction environments.

Response 6: Thank you for pointing this out. The authors agree with the comment. Therefore, we have included in the conclusion: “This specific application extends widely to radiological measurements to the accountancy in deuterium-tritium (DT) fusion energy as a non-invasive gas measurement technique.” This is preceded by “Future applications beyond monitoring CO2 methanation and NH3 cracking experiments include tracking hydrogen isotope exchange in ammonia and methane, first in a “cold” environment using H2 and D2, then eventually in a radiological or “hot” glovebox in the Tritium Instrument Demonstration System at the Savannah River Site.”

Comment 7: It is suggested that the author further clarify the advantages of this technique compared with traditional Raman spectroscopy enhancement techniques (such as surface enhanced Raman scattering, SERS), especially in the aspects of sample complexity and sensitivity improvement.

Response 7: Thank you for pointing this out. The authors FULLY agree with the comment. However, we have not included the enhancement factors in this Communication with the limited amount of data collected on multiple waveguides of the same length in addition to the longitudinal use across waveguides of the same length. A more exhaustive investigation of the characterizing material compatibility of metal coatings for use in a tritium facility is currently funded and just starting this fiscal year. Private communication with Tritium Laboratory Karlsruhe for this particular application. Compared to conventional Raman scattering geometries, the hollow core waveguide can provide sensitivity enhancements of at least 50x. (Consider the reported limits of detection for H2 measured in a waveguide (~100 ppm [11]) and in free space (~0.5% [32]).)

Comment 8: In this study, it is mentioned that the chemometric model is used to quantify the mixed isotopes in real time, and it is suggested to add more details about the model construction.

Response 8: Thank you for pointing this out. The authors agree with the comment. Therefore, we have removed the statement that “models were developed”. For this work, we have stated “This work demonstrates the feasibility to monitor reactive systems by waveguide-enhanced Raman scattering and does not aim to be a comprehensive study on quantifying limits of detection or other figures of merit for a validated analytical method. Further development of instrumental and procedural controls to facilitate operations by more quantitative approaches with lead to the employment of real-time monitoring by Raman spectroscopy.”

Reviewer 3 Report

Comments and Suggestions for Authors

Manuscript ID: sensors-3232931

Title: Online Monitoring of Catalytic Processes by Fiber-Enhanced Raman Spectroscopy

Authors: John T. Kelly, Christopher J. Koch, Robert Lascola, and Tyler Guin

This paper demonstrates the capabilities of a flow cell with a hollow-core waveguide for online measurements of catalytic cracking products. The authors study the methanation of CO2 and the cracking of ammonia as model reactions and use mass spectrometry data as a benchmark. Overall, the work is very interesting and well presented. 

Comments to the Authors:

1)     While it is clear that the Raman intensity-based measurements follow somewhat the MS measurements on Figure 6, there are notable differences in curve shapes and concentrations. What could be the reasoning for this and what could serve as a better comparative method?

Author Response

Comment 1: While it is clear that the Raman intensity-based measurements follow somewhat the MS measurements on Figure 6, there are notable differences in curve shapes and concentrations. What could be the reasoning for this and what could serve as a better comparative method?

Response 1: Thank you for pointing this out. The authors agree with the comment. The following text is added to the text in the manuscript discussing Figure 6: “Subtle difference in the curvature of the Raman and mass spectra are expected as a result from the heated transfer capillary and low flow to the mass spectrometer in addition to the location of the sampling position.”

Round 2

Reviewer 1 Report

Comments and Suggestions for Authors

I have no further comments